# The Expression Patterns of Exogenous Plant miRNAs in Chickens

**DOI:** 10.3390/genes14030760

**Published:** 2023-03-21

**Authors:** Hao Li, Pu Zhang, Diyan Li, Binlong Chen, Jing Li, Tao Wang

**Affiliations:** 1College of Animal Science, Xichang University, Xichang 615013, China; 2College of Animal Science and Technology, Sichuan Agricultural University, Chengdu 611130, China; 3School of Pharmacy, Chengdu University, Chengdu 610106, China; 4College of Agricultural and Life Sciences, Kunming University, Kunming 650214, China

**Keywords:** chicken, plant miRNAs, expression, *miR2018*

## Abstract

(1) Background: MicroRNAs (miRNAs) are involved in a variety of biological processes, such as cell proliferation, cell differentiation, and organ development. Recent studies have shown that plant miRNAs may enter the diet and play physiological and/or pathophysiological roles in human health and disease; however, little is known about plant miRNAs in chickens. (2) Methods: Here, we analyzed miRNA sequencing data, with the use of five Chinese native chicken breeds and six different tissues (heart, liver, spleen, lung, kidney, and leg muscle), and used Illumina sequencing to detect the expression of plant miRNAs in the pectoralis muscles at fourteen developmental stages of Tibetan chickens. (3) Results: The results showed that plant miRNAs are detectable in multiple tissues and organs in different chicken breeds. Surprisingly, we found that plant miRNAs, such as *tae-miR2018*, were detectable in free-range Tibetan chicken embryos at different stages. The results of gavage feeding experiments also showed that synthetic *tae-miR2018* was detectable in caged Tibetan chickens after ingestion. The analysis of *tae-miR2018* showed that its target genes were related to skeletal muscle organ development, regulation of mesodermal cell fate specification, growth factor activity, negative regulation of the cell cycle, and regulation of growth, indicating that exogenous miRNA may regulate the development of chicken embryos. Further cell cultures and exogenous miRNA uptake assay experiments showed that synthetic wheat *miR2018* can be absorbed by chicken myoblasts. (4) Conclusions: Our study found that chickens can absorb and deposit plant miRNAs in various tissues and organs. The plant miRNAs detected in embryos may be involved in the development of chicken embryos.

## 1. Introduction

MicroRNAs (miRNAs) are small, non-coding RNAs which have a vital role in the post-transcriptional mechanism by guiding protein-RNA complexes toward mRNAs [1]. Studies showed that both plant and animal miRNAs are involved in essential roles in development, aging and disease, and the shaping of the transcriptome of many cell types [2,3,4,5]. Some studies indicate that a part of miRNAs can exchange between host and parasites and regulate exogenous gene expression [6,7,8,9]. These reports showed the possibility of cross-species regulation by miRNA. In addition, plant miRNAs are very common and have been detected in the sera, feces, and tissues of animals [10]. Recent studies have shown that mature plant miRNAs can be absorbed through mammalian digestion and perform cross-species gene regulation [11,12], such as maize-derived miRNAs which could be detected in porcine tissues and serum after feeding fresh maize to pigs [12]. Moreover, these exogenous plant miRNAs in the maternal system can even be transferred to the fetus and directly regulate fetal gene expression [13]. Plant miRNAs show uncontrolled dissemination after they are absorbed into animals’ bodies. These reports demonstrate that the cross-species regulation of plant miRNAs is a widespread phenomenon.

Food-derived plant miR168a is found in the sera of mammals and regulates the expression of low-density lipoprotein receptor adapter protein 1 (LDLRAP1) in the liver [14]. *MiR159* is abundant in broccoli and can inhibit cancer growth in mammals by targeting transcription factor 7 (TCF7) [15]. Fungal hyphae can be specifically silenced by miR166 and miR159, which are from cotton plants as the response to infection with Verticillium dahlia [16]. The C. campestris miRNAs may act as virulence factors and regulate host-gene expression during parasitism [17]. A honeysuckle encodes the miRNA, *miR2911*, which can directly target influenza A viruses in vitro and in vivo [18]. The plant miRNAs in beebread can decrease body and ovary size in bee larvae, thereby delaying larval metamorphosis development and inducing larvae development into worker bees [19]. Food-derived ginger exosome-like nanoparticles can be taken up by the mice gut microbiota (*Lactobacillaceae*) and contain ginger miRNAs that target various genes in *Lactobacillus rhamnosus* [20]. However, the expression and potential function of plant miRNAs in chickens has not been explored yet.

Free-range is a term that refers to a method of animal husbandry where farm animals are allowed to move around outside and are not kept in cages; it is a more animal-friendly way of farming. Especially in China, native chicken breeds are adapted to free-range breeding, and long-term farming civilization has also retained this native breeding method. Furthermore, chickens can be raised in the mountains, woods, and grasslands. Free-range chickens are assumed to consume low to moderate levels of pasture, which improves intestinal microbiome and health, and microbial cellulases and hemicellulases may improve the nutritive value of cereal-based diets [21,22]. The long-term intake of plants and even Chinese herbal medicines promote the utilization of probiotics or prebiotics for digestive tract health [23,24]. On a more microscopic level, plant miRNAs in food enter the body to improve the healthy growth and meat flavor of chickens.

In this study, we focused on free-range chickens whose foraging behavior is unrestricted. We collected and examined various tissues and organs of five Chinese native free-range chicken breeds and detected the expression of plant miRNAs through comparison to the mature miRNAs of four different common edible plants, which were wheat, rice, maize, and soybean. The results showed that exogenous plant miRNAs were detectable in multiple tissues and organs of these free-range chickens. Even in embryos, plant miRNAs, such as *tae-miR2018*, were also detected to have a high expression level. Functional analysis of *tae-miR2018* showed that it might be involved in chicken embryo development regulation.

## 2. Materials and Methods

### 2.1. Sample Preparation and Small RNA Sequencing

A total of 20 pectoralis samples were collected from 7 development stages (at the age of 1 day, 36 days, 100 days, 2 years, 5 years, 8 years, and 12 years, respectively) from free-range Tibetan chicken. Meanwhile, we collected 21 free-range Tibetan chicken pectoralis embryo samples, which included 7 embryo stages (chick embryos at 5, 7, 9, 12, 15, 18, and 20 days of incubation, respectively). Except for the 12-year-old Tibetan chickens, which only consisted of two individuals, the other stages all had three replicates. These chickens, raised in the open woodland, consume food freely from their environment and receive a supplementary feed in the form of grains, such as rice, corn, and soybeans. In addition, a lot of fixed sinks were placed to provide clean drinking water. The total RNA was isolated from pectoralis samples by the standard TRIzol method [25] and sequenced on the Illumina HiSeq 2500 (Illumina Inc.; San Diego, CA, USA). To explore plant-derived miRNAs in different chicken breeds, we also downloaded the miRNA transcriptome data of different breeds of free-range chickens from NCBI. The sample information used in this study is listed in Table 1.

### 2.2. Analysis of Illumina Sequencing Data

Fifty miRNA-seq data from five Chinese native free-range chicken breeds and twenty-four miRNA-seq data of ROSS308 were also downloaded from the National Center for Biotechnology Information (NCBI database, https://www.ncbi.nlm.nih.gov/ (accessed on 10 March 2020)) for analysis. The sample information was listed in Table 1. All samples were sequenced on Illumina HiSeq 2500. A total of 115 samples were used for subsequent analysis. Trim Galore (http://www.bioinformatics.babraham.ac.uk/projects/trim_galore/ (accessed on 20 March 2020)) was employed to remove the adapter and low-quality reads (trim_galore -output_dir Trim_galore_out --phred33 --length 20 --quality 20 --stringency 1 seq.fasq.gz). The miRNA detection was performed through Bowtie [26]. We had made two mapping works. The clean data was first to alignment with mature chicken miRNA data that was downloaded from miRbase (http://www.mirbase.org/ (accessed on 23 March 2020)). Then the unmapped reads were mapped to the mature miRNA data of 4 different common edible plant organisms, *Triticum aestivum* (tae), *Oryza sativa* (osa), *Zea mays* (zma), and *Glycine max* (gma), which were also download from miRbase. RNAhybrid [27] was used to do a prediction of miRNA target genes with the main screening condition e < −25 kcal/mol and *p* value < 0.01. In consideration of the frequent noncanonical regulation of miRNA–mRNA [28] and the uncertainty of the function of plant miRNAs in animals’ bodies, we mapped the seed sequences of plant miRNAs to the integrated mRNA sequences of chickens that were downloaded from UCSC (http://hgdownload.soe.ucsc.edu/downloads.html (accessed on 25 March 2020)). The target gene set enrichment was performed with metascape [29]. Because there is no chicken database in metascape, we mapped the target genes to human homologous genes for enrichment analysis.

### 2.3. Animals and Gavage Feeding

The experimental animals were Tibetan chickens, a native poultry breed, with a small body size of a type this is unique to the Qinghai–Tibet Plateau. The bird is distributed at an altitude of 2200–4100 in semi-agricultural and semi-pastoral areas. Animal experiments were carried out in compliance with animal care protocols and all efforts were made to minimize suffering. The protocol was approved by the Institutional Animal Care, and the Use Committee (IACUC) of Sichuan Agricultural University approved the study. Chickens were cage-reared and fed with compound feed with at least 16.5% crude protein and 3.5% crude fat and at most 6.5% crude fiber and 7.4% total ash. Furthermore, it contained 0.9–1.8% calcium, 45% soybean meal, and 0.50% vitamin premix. All diets were treated with high-temperature puffing, and the water was boiled at a high temperature to ensure the experimental chickens were not exposed to exogenous plant miRNAs. The chickens at 42 days of age were fed synthetic miR2018 (1 nmol/kg) by gavage after fasting overnight. After a fixed time interval (i.e., 1 h, 2 h, 3 h, and 4 h), three chickens were euthanized directly. Two tissues (liver and pectoralis) were collected, and the total RNA was extracted. To determine the level of miR2018 in each sample, qRT-PCR was used.

### 2.4. Cell Culture and Exogenous miRNA Uptake Assay in Cultured Cells

Chicken myoblasts were collected from the 10th day of the chicken embryonic stage and cultured in DMEM/F12 (Hyclone, Cat. No. SH30023.01), supplemented with 15% FBS and 1% penicillin–streptomycin at 37 °C with 5% CO_2_. After the second passage, the myoblasts were exposed to a medium with synthetic miR2018 (40 pmol/mL) for different periods of time (i.e., 1 h, 2 h, 3 h, and 4 h). Before collecting the cells, the cells were incubated with a medium that was FBS-free and contained 0.2 mg/mL RNase A for 1 h to digest the extracellularly attached miRNAs. Then the cells were collected and used for qPCR.

## 3. Results

### 3.1. Plant miRNAs Are Detected in Multiple Tissues and Organs of Free-Range Chickens

Through the investigation of the small RNA expression profiles in various tissues and organs, we found that exogenous plant miRNAs were widely observed in free-range chickens. The correlation of expression patterns among all samples in this study showed samples from the same tissue or breed exhibited a similar expression map (Figure 1a). A total of 3268 plant miRNAs with at least one read presented in the sample, including 2656 rice miRNAs, 278 soybean miRNAs, 251 maize miRNAs, and 83 wheat miRNAs, were identified. Next, we investigated the distribution of plant miRNAs in each sample and found that several plant miRNAs were ubiquitous within the selected samples. Illumina sequencing revealed that 15 known plant miRNAs were detected with more than 100 reads in more than 10% of the chicken samples (Figure 1b). Among them, 14 plant miRNAs were identified from *O. sativa* (rice), and the remaining miRNAs were from *T. aestivum* (wheat). These plant miRNAs were widely present in various tissues and organs.

For the Tibetan chickens and the Qingyuan chickens, there were six tissues and organs from each chicken, and we found that the reads of the fifteen plant miRNAs were much higher in the liver than in the other tissues and organs (Figure 1). In the livers of the Qingyuan chickens, up to 5000 reads of *rice-miRf10420-akr* were found; this miRNA exhibited the highest abundance among the detected plant miRNAs and was more abundant than most of the chicken miRNAs detected within the samples. In addition to rice-*miRf10420-akr*, we focused on the miRNA *wheat miR2018* (*tae-miR2018*). It has been reported that *tae-miR2018* can be detected in human serum [30], and up to 1500 reads were detected in liver samples of the Lushi chickens.

### 3.2. Plant miRNAs Are Present in Chickens of Different Ages

To explore the expression of plant miRNAs in chickens across different ages, we collected 41 muscle samples of free-range Tibetan chickens, 21 embryo samples, and 20 post-hatch samples. The samples involved 14 chicken developmental stages. As expected, exogenous plant miRNAs were widely distributed in the muscle of Tibetan chickens. A total of 1033 plant miRNAs, including 760 rice miRNAs, 122 soybean miRNAs, 114 maize miRNAs, and 37 wheat miRNAs, with at least one read present in the sample were identified.

To further identify the influential plant miRNAs, we screened out the miRNAs with more than 40 reads present in the sample and, finally, obtained 33 plant miRNAs, mainly including members of the *miR159* and *miR166* families, *osa-miRf10267-akr*, *osa-miRf11138-akr*, *osa-miRf11479-akr*, and *tae-miR2018*. Among these, *miR159* and *miR166* have been found in mammalian serum [14,15]. Meanwhile, *miR159* has proven to have a function in regulating mammalian gene expressions [15]. Furthermore, *tae-miR2018* has also been found in human serum and was ubiquitously observed in free-range chicken tissues.

Most surprisingly, plant miRNAs such as *tae-miR2018* and *osa-miRf11479-akr* were detected in the embryos of Tibetan chickens and showed regular expression patterns between embryo samples and post-hatch samples (Figure 2a). Among all the detected plant miRNAs, *tae-miR2018* and *osa-miRf11479-akr* had relatively high abundance and were detected in all free-range chickens. Up to 40 copies of *tae-miR2018* were found in 27 Tibetan chicken samples. Moreover, up to 40 copies of *osa-miRf11479-akr* were found in 19 Tibetan chicken samples, and *osa-miRf11479-akr* had higher read counts in the post-hatch samples than in the embryo samples. In contrast to *osa-miRf11479-akr*, *tae-miR2018* showed the opposite distribution and had higher read counts in the embryo samples, and *tae-miR2018* also showed not less than 50 reads in these post-hatch samples.

For comparison, we collected miRNA-seq data for 24 ROSS308 broiler embryos at 4 embryonic development stages that came from a parental line of ROSS308 broilers fed with commercially manufactured feeds. The analysis showed that plant miRNAs were rare or not observed in these samples. In addition, *tae-miR2007* exhibited the highest expression level (only 7 reads) among all the detected plant miRNAs, and *osa-miRf11479-akr* was not detected in the ROSS308 embryo samples (Figure 2b). Only one read of *tae-miR2018* was found in the thirteen-day embryo samples (Figure 2c). The expression levels of these plant miRNAs were too low to be reliable.

### 3.3. Wheat-miR2018 May Regulate the Development of Chicken Embryos

Previous studies [14,15,19] have reported that plant miRNAs present in animal organs are able to perform cross-species RNAi through a similar AGO protein-correlated mechanism. These results indicate that plant miRNAs can regulate animal genes via the high degree of complementarity between miRNAs and target RNAs. Considering the high expression levels of *tae-miR2018* in the free-range Tibetan chicken embryo samples, we next investigated the function of *tae-miR2018*. We first performed bioinformatic analysis to identify any chicken genes matched to *tae-miR2018*. RNAhybrid was employed to predict the target genes of *tae-miR2018*. Metascape was then employed to perform GO and KEGG enrichment analyses of the targeted genes of *tae-miR2018*.

The top 20 enrichment clusters of wheat-miR2018 are shown (Figure 3). These clusters contained 100 GO terms and 3 KEGG pathways. The enrichment clusters were ranked according to significance (significance, *p* value), and the top three clusters were motor learning, skeletal muscle organ development, and regulation of mesodermal cell fate specification. These biological processes are closely related to embryonic development. Forty-two target genes were associated with multiple biological processes, such as bone morphogenetic protein receptor type 1A (*BMPR1A*). *BMPR1A* was involved in skeletal muscle organ development, regulation of mesodermal cell fate specification, sex differentiation, positive regulation of epithelial cell proliferation, regulation of membrane potential, and regulation of growth. This result indicates that *tae-miR2018* may affect the development of chicken embryos.

### 3.4. Synthetic Tae-miR2018 Can Be Absorbed by Chicken Myoblasts and by Chickens In Vivo

To verify whether the exogenous plant miRNAs could be absorbed by chicken tissue in vivo and in vitro, we isolated and cultured myoblasts from the offspring of commercial broiler chickens. After incubation with synthetic *tae-miR2018*, the expression of *tae-miR2018* in the myoblasts rapidly increased and then gradually decreased in a short period of time.

Similar results were obtained from an in vivo experiment: After 12 h of fasting, we assessed the levels of *tae-miR2018* in the livers and pectoralis of chickens fed with synthetic miRNA. The *tae-miR2018* was undetectable in chickens fed normally. We normalized the expression based on the expression level of *U6*. As shown in Figure 4, in the liver, an increased expression level of *tae-miR2018* was detected at 1 h post-gavage feeding. Then, the expression level gradually decreased. In the pectoralis, the expression level showed a trend of first increasing and then decreasing and reached its highest level 2 h after feeding. The highest expression level of *tae-miR2018* in the liver was 5 times higher than that in the pectoralis.

## 4. Discussion

MicroRNAs (miRNAs) comprise a class of 18–25 nucleotide single-stranded, non-coding RNAs that are the smallest known carriers of gene-encoded, post-transcriptional regulatory information in both plants and animals [31]. In the past few years, the debate about whether plant miRNAs can be passively absorbed by mammals has continued. Previous studies have focused on plant miRNAs in mammals and many studies have shown that plant miRNAs can be detected in mammals; these miRNAs have real potential to modulate host gene expression [32,33,34]. In chickens, these local Chinese native breeds are mainly free-range. They consume food freely from their environment and receive supplementary feed in the form of grains, such as rice, corn, and soybeans, given the assumption that some dietary plant miRNAs can be absorbed after ingestion and accumulate within chicken tissues and organs. However, there is no study on the expression of plant miRNAs in chickens. As shown in this study, we found that the plant miRNAs are widely expressed across different developmental stages, tissues, and local Chinese chicken breeds.

The latest research shows that food-derived miRNA absorption occurs in the mammalian digestive tract, especially the stomach, and that low pH inhibits the activity of RNases to protect food-derived miRNAs [35]. Compared to mammals, chickens have a shorter digestive tract. The digestive tract of chickens is also an acidic or weakly acidic environment, and the pH of the crop, proventriculus, and gizzard are approximately 4 [36]. Therefore, plant miRNAs can easily enter the upper part of the digestive tract of chickens and maintain their integrity. In classical physiology, the major functions of the proventriculus and gizzard are believed to be to break down food through mechanical churning and to secrete hydrochloric acid and pepsin rather than absorbing substances. However, we found that exogenous miRNA can normally pass through the digestive tract of chickens and be detected in the liver and pectoralis. O’Neill et al. indicated that following oral administration of plant miRNA, the hostile environment of the gut posed significant barriers to stability [37]. Zhang et al. reported that orally administered plant miRNAs were present in the sera and tissues of animals, suggesting that they may also be resistant to enzymatic digestion in the gastrointestinal tract (GI), and the packaging into microvesicles (MV) and the methylation of plant miRNA may have a protective effect [14]. Recently, the group identified the stomach as the primary site for dietary microRNA absorption and SIDT1 as an RNA transporter that mediates dietary miRNA absorption in the mammalian stomach, and the stomach’s highly acidic environment is crucial for the SIDT1-dependent absorption of miRNAs [35]. We believe that the pattern of absorption of plant miRNAs in chickens may not be the same as that in mammals and that the intestine could be the main organ for the absorption of plant miRNAs in chickens. However, more in-depth research is needed to prove this hypothesis.

Surprisingly, plant miRNAs were also detected in the pectoralis of Tibetan chicken embryos. The expression level of *tae-miR2018* was increased during the embryonic stage (E1-E15) and was higher than the expression levels in the pectoralis of the chickens after hatching. We believe that plant miRNAs could exist in the egg yolk and be protected in a certain way. As the embryo absorbs nutrients from the egg yolk, the plant miRNAs could be ingested by the embryo. Egg yolk is rich in a variety of lipids [38,39] and provides a steady stream of nutrients for fertilized eggs. Some kinds of nutrients in the egg yolk are synthesized endogenously and transported around the oocyte, such as some fatty acids and lipoproteins. Approximately 60% of the lipids are synthesized by the liver in poultry [40]. Approximately seven days before ovulation, oocytes grow rapidly. A wave of lipid synthesis occurs in the poultry liver under the influence of estrogen. Some lipids are synthesized and transferred to the ovaries [41]. We propose a potential explanation. In the process of egg yolk formation, some plant miRNAs are transported to the ovaries along with fatty acids synthesized by the liver. As the embryo develops, the plant miRNAs in the egg yolk are absorbed and deposited in the chicken embryo. In the middle and late stages of embryonic development, various organs gradually form. The exogenous miRNAs are dispersed among the various organs. A study has shown that exogenous miRNAs in humans and mice can be transferred through the placenta to the fetus [13]. We believe that the embryonic development of free-range chickens may also be affected by similar mechanisms. However, there are still problems related to the half-lives and stability of exogenous miRNAs. Our results showed that the levels of exogenous miRNAs in the liver and pectoralis decreased in a short time. Considering that free-range chickens can easily intake plant miRNAs, the expression patterns of exogenous miRNAs may be different between chickens and mammals.

The predicted function of targeted genes of *tae-miR2018* showed that it might be involved in the developmental regulation of chicken embryos. We further performed *tae-miR2018* transfection experiments in myoblasts and found that the expression level of its target gene, *BMPR1A*, was significantly downregulated when *tae-miR2018* was overexpressed. After the addition of a *tae-miR2018* inhibitor, the expression level of *BMPR1A* rebounded. *BMPR1A* is a member of the BMP family, which usually plays key roles in the development of bone and cartilage [42], and some studies have shown that they can act as cytokines in various biological processes, such as sex differentiation [43], embryonic development [44,45], and angiogenesis [46,47]. As a receptor of BMP signals, BMPR1A binds ligands to form a complex that regulates the expression of downstream genes. Studies have shown that *BMPR1A* is involved in adipogenesis [48], osteogenesis [49], and hair follicle development [50]. In chickens, studies have found that BMP signals affect the inner ear [51] and early eye development [52]. In any case, BMP signals and their receptors have important influences on embryonic development. Further studies should be conducted and explore the effects of exogenous plant miRNAs on the development of chicken embryos. In total, here we found that plant miRNAs can be ingested by chickens and deposited into various tissues and organs. More importantly, they can be transferred to chicken embryos and may affect the development of chicken embryos. These findings will provide insights into the regulatory role of plant miRNAs in the growth and development of domesticated animals.

## 5. Conclusions

In this study, using chicken as an animal model, we investigated whether there are plant miRNAs in birds. Through bioinformatic analysis, we found that plant miRNAs can be detected in multiple tissues and organs of multiple types in different chicken breeds, such as Tibetan chickens, Qingyuan chickens, Gushi chickens, Lushi chickens, and Xinhua chickens. Some plant miRNAs were detected with thousands of reads. In addition, plant miRNAs were also found in the embryos of Tibetan chickens. The deposition levels of wheat *tae-miR2018* in the embryonic tissues were even higher than those in the pectoralis of free-range chickens. In vitro cell culture experiments also showed that *tae-miR2018* could be absorbed by myoblasts. In summary, our research expanded the existing information on cross-kingdom regulation by exogenous plant miRNAs, and further studies need to be conducted to explore the regulatory function of these plant miRNAs in animals.

## Figures and Tables

**Figure 1 genes-14-00760-f001:**
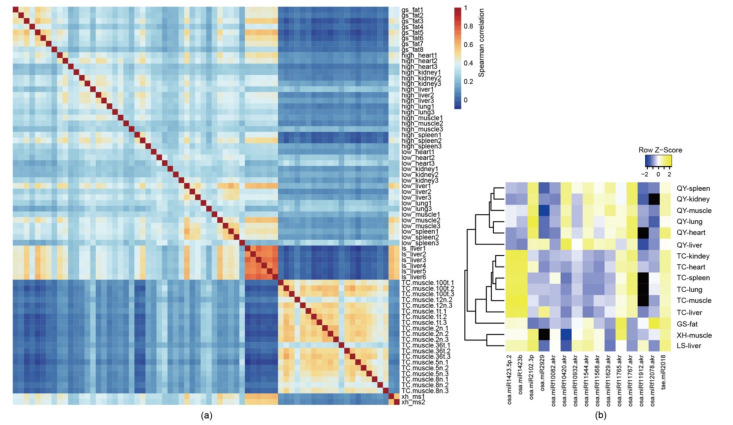
Expression scheme of plant miRNAs in different chicken breeds and tissues. (**a**) Spearman’s correlation heatmap of the plant miRNA expression among different chicken breeds and tissues. (**b**) Mean expression levels (reads counts) of 15 plant miRNAs were detected in 7 tissues or organs from 5 Chinese native free-range chicken breeds. Note: *O. sativa* (rice, osa), *T. aestivum* (wheat, tae).

**Figure 2 genes-14-00760-f002:**
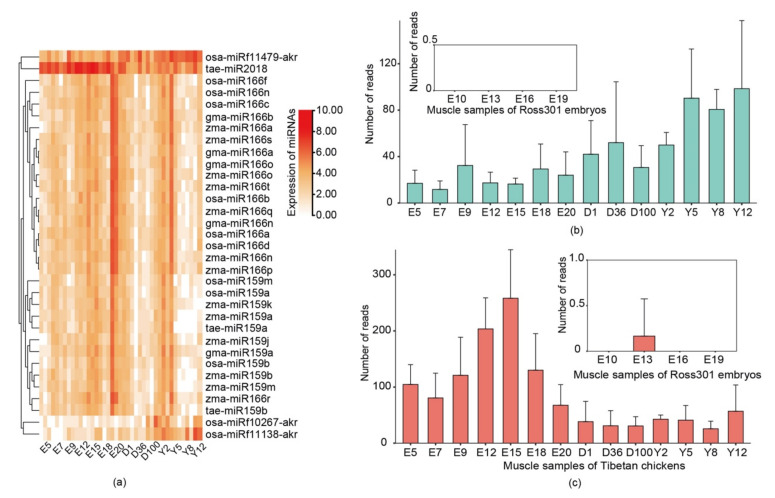
Expression of plant miRNAs across different chicken developmental stages. (**a**) Heatmap showing the expression of plant miRNAs with more than 40 reads. Expression levels (read counts) of *osa-miRf11479-akr* (**b**) and *tae-miR2018* (**c**) at 14 chicken developmental stages. Note: ‘E’, ‘D’, and ‘Y’ indicate the sampling times in ‘embryonic days’, ‘days after hatching’ and ‘years’, respectively. Expression of plant miRNAs in Ross301 chicken feed without exogenous plant miRNA is also shown.

**Figure 3 genes-14-00760-f003:**
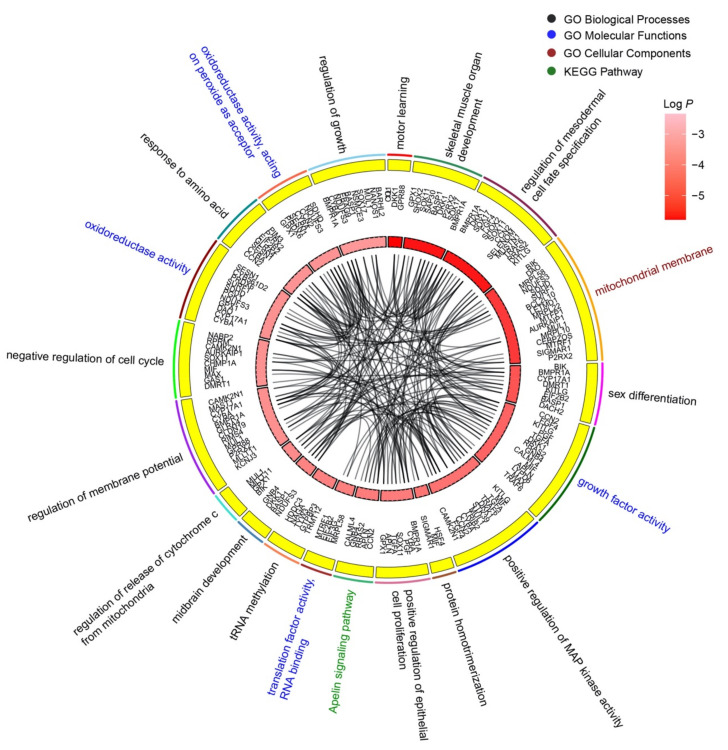
Enrichment analysis of target genes of *tae-miR2018*. The top 20 enrichment clusters are shown. The genes in each cluster are marked in the second circle. The third circle shows the correlation degrees of the target genes and the pathways arranged by *p* value. The connections in the chord diagram indicate that the corresponding gene is associated with multiple pathways and GO terms.

**Figure 4 genes-14-00760-f004:**
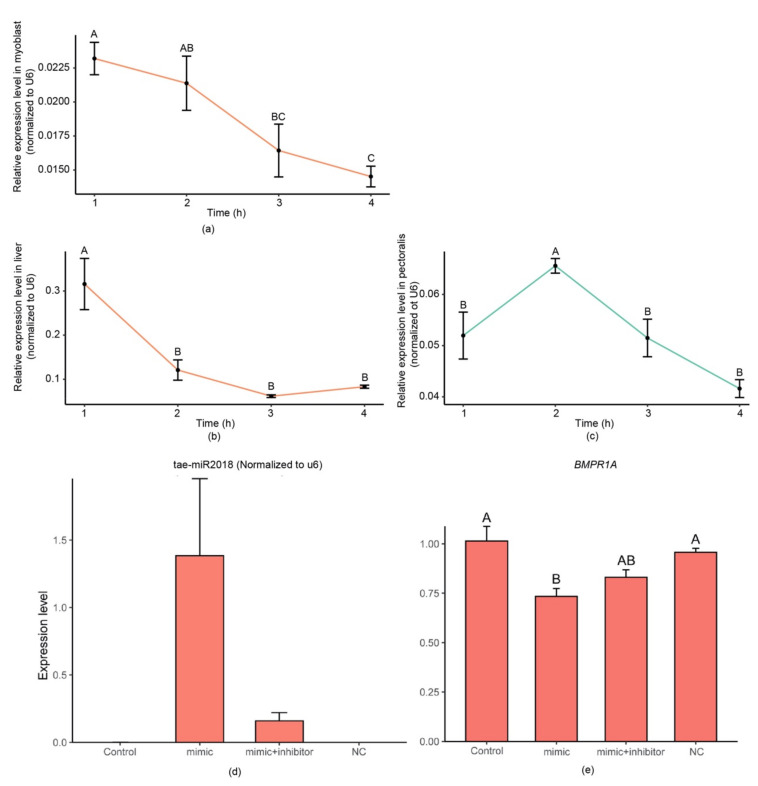
Expression levels of *tae-miR2018*. Relative expression levels of *tae-miR2018* in myoblasts (**a**) after incubation with synthetic tae-miR2018 and in the liver (**b**) or pectoralis (**c**) of chickens at the indicated time points after gavage feeding of synthetic dietary miRNAs. One-way ANOVA with the Tukey method was used to evaluate significance (*p* < 0.001); ‘h’ means hour. Expression levels of tae-miR2018 (**d**) and *BMPR1A* (**e**) in chicken myoblasts transfected with wild type, *tae-miR2018* mimic, *tae-miR2018* mimic, and inhibitor ncRNA constructs.

**Table 1 genes-14-00760-t001:** The summaries of sample information.

Chicken Breed	Tissues	Number	NCBI Access Number	Age
Tibetan chicken (TC)	Heart	3	PRJNA511987	>200 days
	Liver	3		>200 days
	Spleen	3		>200 days
	Lung	2		>200 days
	Kidney	3		>200 days
	Leg muscle (Tibialis anterior)	3		>200 days
Qingyuan chicken (QY)	Heart	3	PRJNA511987	>200 days
	Liver	3		>200 days
	Spleen	3		>200 days
	Lung	2		>200 days
	Kidney	3		>200 days
	Leg muscle (Tibialis anterior)	3		>200 days
Xinghua chicken (XH)	Pectoralis muscle	2	PRJNA266323	7 Weeks
Gushi chicken (GS)	Fat (Abdominal adipose tissue)	4	PRJNA528858	6, 14, 22 and 30 weeks
	Pectoralis muscle	4	PRJNA516961	6, 14, 22 and 30 weeks
Lushi chicken (LS)	Liver	6	PRJNA299589	20 and 30 weeks
Tibetan chicken (TC) ^1^	Pectoralis muscle	21	PRJNA699998	Embryo stage (5, 7, 9, 12, 15, 18 and 20 days)
		20	PRJNA699998	After hatching (1, 36, 100 days and 2, 5, 8 and 12 years)
Broiler (ROSS308)	skeletal muscle	24	PRJNA516545	Embryo stage (10, 13, 16 and 19 days)

^1^ Represented data generated in this study. Data of other breeds were downloaded from NCBI. The Gushi chicken is a domestic Chinese breed in Henan, and it is often used for breeding and production. Lushi chicken is one of the three best breeders in Henan. It is an ancient chicken breed suitable for mountain stocking. Lushi chickens are light, sturdy, and resistant to rough feeding. The Qingyuan and Xinghua chickens are indigenous chicken breeds in Guangdong that grow in beautiful mountains. Mainly, it is the mountain and forest chicken, and, in the mountain forest, slopes run all day to feed.

## Data Availability

The high-throughput sequencing data of RNA-seq have been saved in the NCBI Sequence Reading Archive, with the accession number PRJNA699998.

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
