# Peer review of "The Expression Patterns of Exogenous Plant miRNAs in Chickens"

_genes, 2023, doi:10.3390/genes14030760_

Round 1

Reviewer 1 Report

Dear Authors,

The presented study analyzes the increasingly current and fashionable topic of miRNA and the

possibility of interactions between organisms in terms of its presence, etc. While the topic is

interesting, the methodology used in the presented form, the tools used to analyze some issues raise

doubts. In the case of slow-growing hens, there is no detailed description of the habitat of birds,

which makes it impossible to clearly indicate what causes the presence or absence of certain

miRNAs in chicken tissues. Information on nutrition is also rudimentary, we do not know to what

extent their feed was supplemented and with what additives, such as cereals, etc. (amount, time of

administration, etc.). This means that we do not know to what extent a particular feed affects the

bird's body and is a source of plant miRNAs.

After reading the presented text, analyzing the cited publications and those available from similar

studies, some questions arise.

1.     What was the criteria for the breeds selection? Why Ross 308 as a typical 42days old broiler (fast growing) was compared to the slow growing bred? Why any of standard slow growing bred (Ranger Classic, Ranger Gold, Rustic Gold, etc.) was chosen for the comparison?

2.     Why all the same types of tissues were not collected for all the breeds? Only for TC and QY chicken there all the same combination of the type of samples. All the rest have a single tissue sample. The full, real comparison can be correctly made only between TC and QY.

3.     What does it mean leg muscle? Fat- what kind of fat? Please, give more details about samples.

4.     The chickens eat food scraps, grain and what- ever else they forage. Based on this information we cannot be sure, what element of the feed or anything different (like drinking water etc.) may affect the miRNA expression. Please, give more details about the experiment environment, to be sure that the presence or absence of analysed miRNAs is related to the feed factor not to the others. Based on the presented data we cannot be sure that only plants can distribute these small molecules into the chicken body.

5.     Total RNA was isolated from all pectoralis samples by the standard TRIzol method- please give the details or references.

6.     Trim Galore (http://www.bio-88 informatics.babraham.ac.uk/projects/trim_galore/) was employed to remove the adapter and low-quality reads. What was the final criteria for the reads use for analysis?

7.     Why the authors chose 3 plants, what was the criteria to choose this kind not the others?

8.     The target gene set enrichment was performed with metascape [24]. Because there is no chicken database in metascape, we map the target genes to human homologous genes for enrichment analysis. There are other tools like targetscan where there is a possibility to make the analysis for different species, also for Gallus Gallus.

9.     Please, provide the full data about the feed program and feed composition that was used in the experiment (2.3)

10.  L115- should be DMEM

11.  L 239-242, please remove this (it is a part of the instruction for authors, there is no need to have it in the original paper)

12.  There is a lack of info about the parental flocks for the slow-growing birds, especially in the feed aspects (there is an info about Ross 308 parental flocks and type of feed (described as commercial)).

13.  L328 prepared instead of prepare

Hope, this will be helpful for you to improve you manuscript.

Reviewer 2 Report

Abstract: The abstract section was well .,

In L#15-L#16 In the abstract section, it will be best to write the material method in just two lines. Please  discuss how five breeds of native chickens were further divided into groups or replicated before  L#15. Must be mentioned the name of the tissue or organs in  L#16. In the result section authors never mention the group in which miRNAs expresses in L#18-L#23.

Introduction: introduction summarizes relevant information. Add  one paragraph at the start of the introduction section that why we used plant/herbal/prebiotic or phytobiotic in the diet of broiler.

Materials and Methods: The methods are adequate. But in the 1st  paragraphs, L#69-L#79 difficult to understand the mean number of birds or their division is ambiguous and can’t be understood.

Discussion: Discussion-(Intellectual vigor, logical interpretation of results, and alignment of conclusion with the results, etc.).

In the discussion section please discuss the architecture of the intestine and how absorbed the miRNA and their retention in tissue.L#256-268.

References: Please add references according to the journal format. Some use the abbreviation journal and in others use full names..

Round 2

Reviewer 1 Report

Dear Authors,

The revised version looks much better and mostly clears up the previously problematic issues. Completed descriptions clarify your breed choices and methodology.

However, in my opinion, in a feed experiment, it is absolutely necessary to provide the full composition of the feed mixture in the form that is most often presented in publications of this type, e.g. Effect of Antibiotic, Phytobiotic and Probiotic Supplementation on Growth, Blood Indices and Intestine Health in Broiler Chicks Challenged with Clostridium perfringens,Licorice Extract Supplementation Affects Antioxidant Activity, Growth-Related Genes, Lipid Metabolism, and Immune Markers in Broiler Chickens,The Effect of Administration of a Phytobiotic Containing Cinnamon Oil and Citric Acid on the Metabolism, Immunity, and Growth Performance of Broiler Chickens, also to demonstrate that the diet is a standard diet and is not an experimental factor that could influence the results.

GO for target genes, as I have already indicated, should be analyzed using programs with the option of selecting species specificity - if available. The ability to choose the species Gallus Gallus gives, for example, DAVID or Panther.
